# Health experiences of pregnant and women with chagas disease in the Atacama, Tarapaca, and Metropolitan regions of Chile. Mistreatment as an indicator of healthcare barriers

**Andrea Avaria**[1][ORCID]*, **Camila Plaza**[2]

**1** Facultad de Ciencias Sociales y Humanidades, Doctorado en Ciencias Sociales, Universidad Autónoma de Chile, Santiago de Chile, Chile, **2** School Museum studies, University of Leicester, Leicester, United Kingdom

☯ These authors contributed equally to this work.

\* andrea.avaria@uautonoma.cl

**Data Availability Statement:** Data are available from Ethics Committee (contact via comite. etica@uautonoma.cl) for researchers who meet the

## Abstract

### Introduction

Congenital transmission is one of the most significant forms of *Trypanosoma cruzi* transmission worldwide. Migrant women, in particular, often face barriers to accessing the healthcare system; one such barrier being that their health rights are not recognised. The situation in Chile is a reality that can be extrapolated to historical territories affected by Chagas disease and that is characterized by migrant populations. This article explores the healthcare experiences of pregnant and women living with Chagas disease, both nationals and migrants, and residents of three regions of Chile.

### Methodology

The qualitative research study analyzed the experiences and meanings around the problem of Chagas based on 54 in-depth interviews with women in pregnancy and women who were diagnosed with the disease. The information was processed following the Grounded Theory tenets through the constant encoding feedback procedure, which allowed us to describe and comprehensively understand the phenomenon.

### Results

The findings indicate that managing information is a sensitive aspect in evaluating healthcare experiences, with the potential to either positively or negatively impact the acceptance of diagnosis and treatment planning. The negative impact is determined by the communicative dimension and how healthcare teams deliver information. Poor attention, poor treatment, and lack of empathy lead patients to reject or distance themseves from the healthcare system. The positive aspects are related to a sensitive, personalized, and highly empathetic treatment. In historically endemic areas, these factors are essential for ensuring continuity

criteria for access to confidential data. The Ethics Committee of the Universidad Autónoma de Chile, as the institution responsible for the execution of the project, undertakes to make the results available to those who request them in accordance with access criteria that are consistent with the principles of public use and common good.

**Funding:** This research was funded by Fondo nacional de investigación y desarrollo en Salud, Agencia Nacional de Investigación y desarrollo, Ministerio de ciencia, tecnología, conocimiento e innovación, grant number FONIS SA18I0056.

**Competing interests:** I have read the Journal´s policy and the authors of the manuscrip have the following pompeting interests. The authors declare no conflict of interest. The funders had no role in the design of the study; in the collection, analyses, or interpretation of data; in the writing of the manuscript; or in the decision to publish the results.

in healthcare processes. There is an evident need to understand and value the settings, means, and contents within Chagas´s healthcare contexts.

## Conclusion

Communication is crucial during diagnostic processes as it determines the assessment, credibility, and trust in the health system, thereby influencing the continuity of treatment, especially in highly sensitive moments such as pregnancy. Therefore, it is necessary to articulate strategies considering information magement with greater empathy and emotional support.

## Introduction

The World Health Organization (WHO) estimates that about eight million people are infected in the endemic area of Chagas disease, mainly in the 21 continental countries of Latin America [1]. According to information from the Chilean Ministry of Health, Chile—declared free of home vector transmission by *Triatoma infestans* in 1999 and recertified in 2016—has an estimated 120,000 people diagnoses positive for *Trypanosoma cruzi* parasite, which causes the disease. Human mobility has revealed the enduring presence of Chagas as a parasitic disease, constituting a global public health issue. This has become evident in countries where it was previously unknown and has made it relevant in historical endemic countries.

Since 2016, Chile has been declared free of vector transmission and maintains strict vigilance in blood and organ donation. However, the national seroprevalence is 1.2% IgG in 2017. The Ministry of Health introduced The National Plan for Chagas Disease in Chile in 2014 through the Health Technical Standard n˚162. The norm prescribes the screening and treating of national and migrant pregnant women, newborns, and blood donors. Since its establishment, it has facilitated the detection, treatment, and accompaniment of people with a positive diagnosis for Chagas disease. Nevertheless, Chagas disease continues to be a significant health problem.

The relevance of establishing strategies aimed at improving adherence to treatment among women diagnosed with this disease underscores the pertinence and significance of researching their assessments of their healthcare experiences. Despite progress at the national and regional levels, it has been reported that many people do not seek adequate healthcare after being diagnosed with the disease [1–3].

The literature shows that one of the leading research concerns related to Chagas disease and migration focuses on the challenges for health systems and the success or failure of specific health programs implemented in regions receiving migrants [4]. Some analyses have focused on the increase of Chagas disease diagnoses among migrants who have migrated to non-endemic countries in Europe, especially Spain, Germany, Italy, and England. In these countries, low adherence to specific programmes developed in some health centres has been highlighted as a contributing factor to the development of diseases resulting from Chagas disease [5–10]. As a second area of investigation, there are studies focused on the persistence of Chagas disease in Latin American countries. These aim to analyse Chagas prevalence in places where the disease is traditionally known but where there is a significant lack of awareness about its risks in people's life trajectories [11–15]. Finally, some investigations address social determinants of health and Chagas disease. Those studies analyse the social and cultural

dimensions, such as the socio-economic, political, and cultural conditions that impact Chagas disease prevalence and transmission [16–19].

Studies that have addressed Chagas disease from the perspective of women are even more scarce. Such studies have highlighted different dimensions of Chagas disease from the women's point of view. Some have highlighted the life experience of Bolivian migrant women with the disease. [11] analysed the perinatal risk of Bolivian migrant mothers and their newborns compared to Argentine mothers and their newborns, considering the prevalence of Chagas disease, among other factors. Arenas-Monreal et al. affirmed that social determinants and the gender approach are imperative to understand the vulnerability of populations, their exposure risks, the determinants of the attention they recieve, and their participation in the prevention and control of vector-borne diseases [20]. They [20] concluded that gender models the risks of exposure to vectors of disease transmission. Finally, a research conducted at the José María Ramos Mejías hospital in Buenos Aires [21], Argentina, aimed at describing the prevalence of Chagas disease in pregnant women. Specifically, it analysed Bolivian migrant women due to the dominance of the disease in their country of origin in order to evaluate the impact of these migratory flows on the "urbanisation" of the disease in Argentina.

Based on the conviction that woven into their experiences and meanings are pivotal elements influencing the acceptance of diagnosis and adherence to treatment, the present study aims to analyse the experience of women diagnosed with Chagas disease through their evaluation of the healthcare they received. This article was developed in the context of project titled "Chagas challenges for today's Chile: diversity, migration, territory, and access to rights. A quantitative and dynamic approach to Chagas disease care in the Tarapacá, Atacama, and Metropolitan regions" which addresses the problem of Chagas disease while recognising the importance of subjective components related to this disease. Recognising the significance of women's subjectivity in the healthcare trajectories of Chagas disease underscores the crucial role of communication management in the processes of diagnosis, treatment, and follow-up, and emphazises the centrality it should have in the design of more empathetic and emotionally-containing communication strategies.

## Materials and methods

This article presents the results of significant qualitative research conducted in accordance with the principles of Grounded Social Theory, as proposed by [22,23]. The objective of this study was to identify the meanings and experiences of women, both nationals and migrants, diagnosed with Chagas disease during pregnancy or later. The project aimed to produce a theoretical formulation of the meanings attributed by women to the Chagas problem and to explain the disease experience through constant linkage with field data collected in three regions of the country.

The information was collected defining the differentiated regional and institutional contexts to be analysed, considering the definition of historically endemic areas and current migrant concentration. Teams were formed in each region and trained on the ethical requirements for conducting interviews in the context of social sciences research for health. The regions were selected considering as criteria the prevalence of the disease in northern Chile and included the primary urban centre of the country, the Metropolitan Region. It should be noted that these regions concentrate a significant number of the migrant population [Fig 1]. This selection allows us to observe possible similarities and differences that may arise from the testimonies.

The participants were selected by the treating healthcare teams at the hospital or primary care level in each region. These teams facilitated the first contact with the women. Following

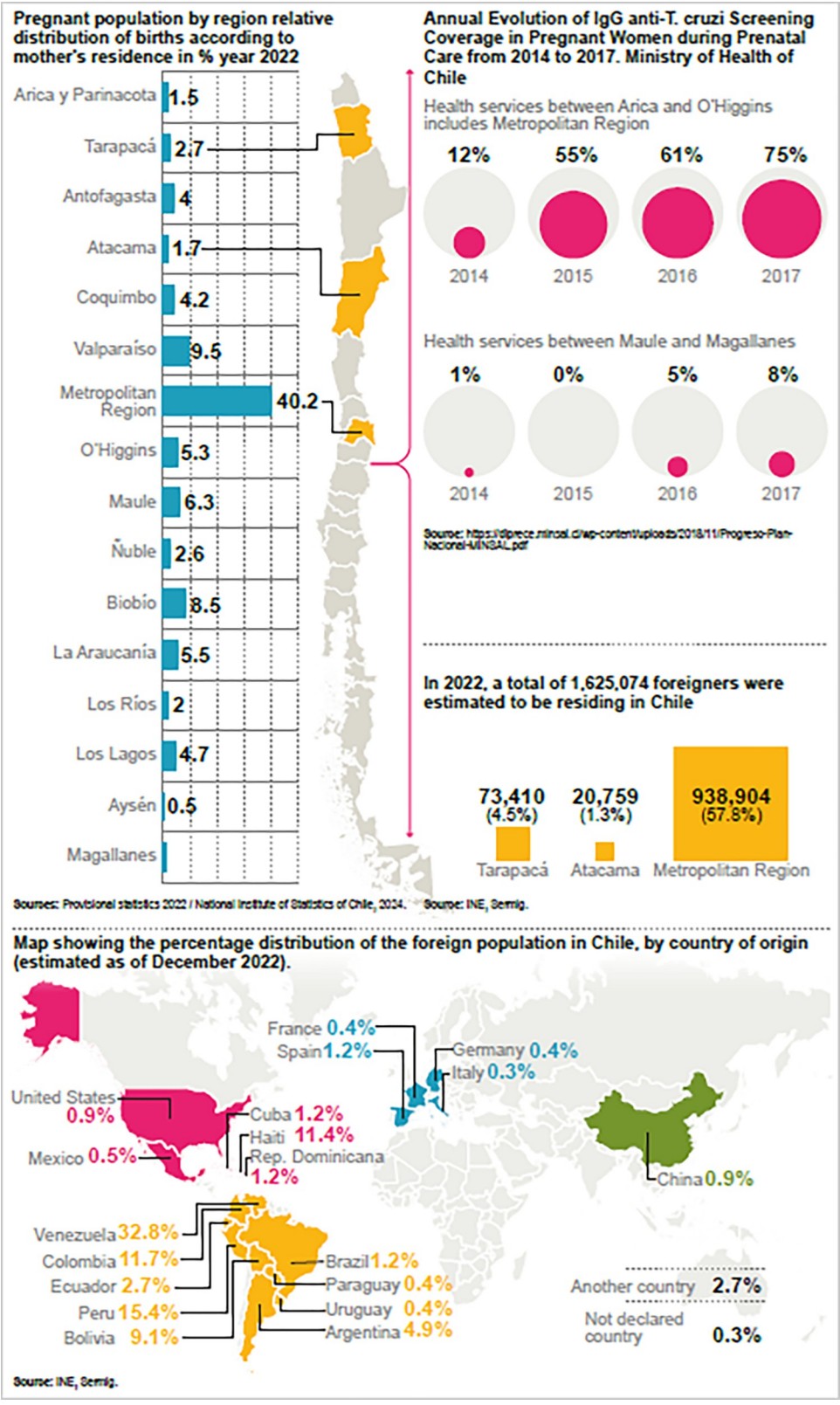

**Fig 1.**

this, the local project teams contacted the women by phone, confirmed their participation, and obtained informed consent. All participants had to have been diagnosed positive for Chagas disease after 2016, when Technical Standard n˚162 was implemented in the Chilean health system.

The in-depth interviews were initially conducted in person, the contacts were made and the data collections begun at september 27, 2019; then transitioned to online formats to comply with health measures and social distancing protocols related to the SARS-CoV-2 pandemic, to december, 30 2021. Available evidence on digital data collection has shown that online interviews do not interfere with the quality of data to be collected [24–26]. The project began its administrative work on January 27, 2019, and administratively closed the project on March 26, 2022.

The questions were defined so that they could provide us with relevant information that would allow us to investigate the meanings they give to their experiences, including their migration trajectories (in the case of migrant women) as well as experiences and meanings regarding the Chagas problem.

The research team recorded the interviews with the prior consent of the participants. A team of research assistants later transcribed them, and procedures were followed to protect the identity and confidentiality of the interviewees.

The interview transcriptions were analysed following the principles of Grounded Social Theory that prescribe open, axial, and secondary coding procedures. The codification process allows for constructing general interpretations of the phenomena studied, using theoretical saturation and constant comparison of the data from the materials during the coding process. Atlas ti 7.0 software was used to facilitate the coding process.

The research complied with the ethical requirements set forth by, initially, the Ethics Committee of the Universidad Autónoma de Chile, and the Ethics Committee of the Western Metropolitan Health Service of Santiago de Chile considered the Declaration of Helsinki of the World Medical Association. The research proposal was ratified through a letter initially from the Ethics Committee UA, CC N˚128–2018, and Ethics Committee, CCE No 18–2019. Given the context of the pandemic, adjustments and necessary changes were made, requiring a new ethical evaluation, which was confirmed through the letter CCEN 19–2021.

## Results

### Characterisation of participants

The study included 54 women living in 3 regions of Chile: Tarapacá and Atacama in the north of the country, and the Metropolitan Region in the center. The participants were 21 to 46 years old; 23 were Chilean, 29 were Bolivian migrants, and 1 argentinian, and 1 from Venezuela. The median age is 32. Among the migrant participants, 5 had university or higher technical education, 14 had complete school education, and 12 had incomplete school education. In total, 14 participants were primiparous, and 40 multiparous. Additionally, 13 were married, 21 were in a relationship, and 20 declared themselves single.

### Assessments on care for Chagas disease

In the experiences shared by the interviewees, there is a clear and defined opinion regarding the healthcare assistance they received once diagnosed with Chagas disease. The participants formed clear judgments about their experiences with the health institutions and the health personnel with whom they interacted. The testimonies highlighted the importance of information management, including its settings, means, ways, manners, and contents, in shaping the positive or negative evaluation of their experiences. A better or worse perception of the healthcare

received since the diagnosis is defined by articulating two crucial elements: information and treatment.

## Negative review

The negative assessment is strongly determined by the communicative dimension and the capacity of guidance and containment that healthcare teams have in transmitting the information. The participants expected these teams could guide and clarify doubts and questions about Chagas disease. They are expected to know the regular channels of information and the appropriate procedures they must follow after the diagnosis.

Interestingly, when examining the distribution of negative quotes according to the territory and nationality of the interviewees, it was observed that Chilean participants and pregnant or women from the Metropolitan Region were the ones who tended to express negative sentiments about the healthcare they received. This could be related to the fact that healthcare teams in the Metropolitan Region face Chagas disease diagnoses less frequently. This lack of familiarity with the disease results in little knowledge about the system and the consequent waste of time accused by the participants even in historically endemic areas, as we see in the following story.

> ¨A lot of confusion. . . paperwork (. . .) horrible organisation, horrible administration. Not only with that, with many tests I underwent—tests that were lost, that they could not find. . . that. . . I had to undergo them again because something had failed in the test-taking process. Because they would schedule a test that required fasting and tell me to go without fasting. . . So, that's why I say that the quality of public health is not the best. . .¨ (Chilean, Atacama Region, 26 years, primiparous, single).

## Poor attention: Bad manners, wrong ways, lack of clarity

For participants, poor healthcare consists mainly of qualitative elements related to the ways and manner in which they were treated and how the healthcare teams delivered the information. Additionally, a bad assessment involves practical issues. The lack of clarity in procedures and the lack of coordination among the various actors and institutional instances involved in the process become essential. Some experiences are echoed across multiple testimones: wandering from one place to another, prolonged and uncertain waiting times for test results, the permanent feeling of wasting time waiting, and discoordination due to the lack of clarity of the institutions and health personnel. Thus, poor attention usually results in pregnant women not being adequately directed or guided.

> "You have that availability, because you want to get cured. When I went there, they sent me to the CESFAM. They did the analysis. They told me, 'Come back in 5 days to receive the results.' After 5 days, the person from the laboratory tells me, 'They're still not ready.' 'When should I come back?' I said. 'Maybe my boss can give me permission.' 'I don't know. Just come back to find out.' So I thought, some Bolivians don't have that availability due to their work, because their bosses sometimes don't want to give them permission. I returned again, another 10 days had passed, and the tests were still not ready. Then I had to travel to Bolivia. I was 2 weeks there, and then I came back again. Before that, I told my boss, 'Every time I go to get the test results, they're not ready, and they don't give me an exact date.' So he says to me, 'I'm not saying this for myself, but I'm saying it for the other people who will

want to go to the doctor'. And if one doesn't have that time available, then you get tired and say, 'Oh well, I'll just leave it.'" (Bolivian, Atacama Region, 46 years, multiparous, single).

Some of the participants also reported ill-treatment in the process. In these cases, openly discriminatory expressions due to their status as migrants, perception of nasty looks, ominous tones when speaking, or harsh treatment when performing health procedures repeat across the negative experiences described.

¨Yes. . . there are times, doctors, nurses, that suddenly. . . Once, they told me why we came here to make the government spend more money or why we won't demand better things from our country. In that case, it happened to me. . . to me. . .¨ (Bolivian, Metropolitan Region, 34 years, multiparous, married).

Another aspect that emerges from these abuses is the perception of having been treated as ¨weird bugs¨. Voice tones of surprise and curiosity on the part of the health personnel produce discomfort among the participants and lead to a negative evaluation of the treatment received.

¨Sure, when it happened to me, I told this to the doctor, but something happened to me when I was hospitalised. Because, I didn't know what was going on, because the doctors, you go in, they look at you, and suddenly around ten doctors come into the room, so you feel like a weird bug because they start talking about you, like speaking among themselves, so you don't understand, the truth is you don't understand¨ (Chilean, Metropolitan Region, 41 years, multiparous, single).

As can be seen, the emotional dimension of treatment and attention is fundamental for the participants, as is the aspect of information. Not only are the scarcity or absence of sufficient explanations assessed negatively, but also the ways, manners, and means by which the teams conveyed these explanations. In this way, receiving information by telephone without further explanation, as well as experiencing long waits and excessive apprehension when the diagnosis is reported, is perceived negatively. The notion that the communication of the diagnosis was done lacking empathy and transmitting fears and insecurities is part of a poor perception of the attention received.

¨I feel that she was like super little empathetic, the girl who informed me. Also understanding that I was pregnant and had just found out, and that I had had another lousy pregnancy experience. She told me that I had to talk to the infectologist, you know? And the procedure, I also want to tell you, is super inappropriate, I believe. Because, in the end, they tell you over the phone that you have to go urgently to talk to the girl over there, so you go and like, okay, I came, and what happens? No, 'You have to wait because the boss has to attend you, because we're not authorised'. . . so, I don't know, you think it's something else entirely. So hear this out. I went into the office and, 'Okay, but you have to be alone, your husband has to leave,' and I was like, 'no, I do not want him to leave". . . so everything was. . . to find out that I had Chagas disease was stressful. I think I had the baby here (points to her neck), you know? Like that. It was terrible. No, no, super bad¨ (Chilean, Metropolitan Region, no age information, multiparous, no civil status information).

¨But there was an episode that happened to me. I think Ana was the girl's name; the truth is I don't even remember anymore. I was hospitalized one night and she looked at me and said, 'Have you asked what you have?' I said, 'No, I haven't asked anything.' And she looked at me and said, 'Yes, because what you have, you can die from it.' So I was plop [shocked],

and then that same girl looked at me and said, 'Why do you have that face?' And I looked at her and said, 'What face do you want me to have after what you just told me?' And I told this to the doctor. And I told her, I gave her an example, I told her, I am a person, I consider myself super strong, that I have been through a lot in my life, so, on top of that, I am a single mom, so one has to be strong. So I told her, if it had been a weaker person mentally, and that person, I had the window right there, if it had been another person they would have killed themselves. So I told her that I thought she should not have done that. And, well, she did it to me, I don't know if she's done it with any other patients, but I told her it's not her, I think it's the doctors, not the nurses [that should convey delicate information]¨ (Chilean, Metropolitan Region, 41 years multiparous, single).

¨And, uh, but my pregnancy, at the beginning, it was terrible because the doctor, who was a lousy doctor in my opinion, I, uh, I mean, she said, 'Everything is wrong,' as if my daughter was going to be born with malformations; that, if I didn't know anything about this, how could I dare to get pregnant with him. And then I said to her, 'No,' I mean, 'I didn't know I had this when I got pregnant.' Anyway, lousy doctor. I left there crying, and I wandered alone, calling my husband on the phone and everything¨ (Chilean, Metropolitan Region, 40 years, multiparous, no civil status information).

The last dimension that contributes to a negative perception in terms of information relates to the participants' perception that the health personnel did not handle the information about the disease correctly and accurately. This uncertainty transmitted by the health personnel causes great insecurity in women during pregnancy. In this way, references to the fact that the health personnel gave them contradictory and confusing information or that they themselves did not know about Chagas disease are echoed among the accounts that evaluate their experiences negatively. This allows us to affirm that the moment in which the information is delivered, during gestation, is crucial. The reception, emotional containment, and treatment at this stage impact people's evaluation of the system; a system that lacks information, transmits insecurity, and reproduces discomfort in people.

"Yes, I think there was a lack of organisation and more information considering how people were operating right there. Because the doctor, when he saw me, said something totally different to what I learned later. Because the doctor told me, 'You can't breastfeed, forget about it. You're going to transmit your Chagas disease and delay your treatment through breastfeeding. So I forbid you to breastfeed; when you have your child, you won't breastfeed.' And I was shocked. I thought, 'Okay, I'm going to have to not breastfeed him.' And, obviously, when you're a mom, you want everything to be perfect, the attachment and all that. Then I left his office and went to the. . . and he said, 'What does the doctor say?' And I told him this, that he forbade me to breastfeed. And he said, 'No. Here in Chile, it's not the same.' Because the doctor, I do not know where he is from, maybe Ecuadorian and. . . 'It is not the same as in his country. You can breastfeed here; there is a law that supports breastfeeding. But it is your decision. If you do not want to give milk and prefer to start this treatment as soon as possible to help the little one, it's up to you to decide'" (Chilean, Atacama Region, 26 years, primiparous, single).

¨So I arrived at the I-do-not know-what, ask if he remembered me, and said, 'Well, here is the result.' And he said, I mean, Gutiérrez, the high-risk chief of the Hospital, the sonographer, I mean, a guy who has to be like fifyt years old, fifty-something, and he tells me, 'I had never seen a pregnant woman with Chagas before.' I thought 'God, what do I do now.'¨ (Chilean, Metropolitan Region, 36 years, multiparous, married).

¨they should have more information. Because I, what I realised was that the doctor who attended us, she would read in order to give us the pills. . . she would read from the internet. I don't know if she was from another country or I don't know, but she would read how much the dosage of the pill was, for how many days I had to take it. . . she didn't know. . .¨ (Chilean, Metropolitan Region, 35 years, multiparous, married).

Finally, the last element contributing to a negative assessment of healthcare is the lack of a satisfactory explanation accompanying the diagnosis. In other words, the perception that they give confusing information that fails to answer their doubts about the disease. In this way, references to the healthcare team not informing them about the test for Chagas disease, providing very rapid, very general, or superficial explanations of the disease after diagnosis, or failing to offer information that would allow them to feel informed are repeated across testimonies.

¨I would be able to explain very little because at least I wasn't given much information; they gave more infromation to my daughter¨ (Chilean, Atacama Region, 36 years, multiparous, single).

¨With the Chagas disease test. . . they didn't tell me they were going to test for Chagas disease. I didn't know what Chagas disease was. They just told me they were going to take a blood test for anaemia and a urine test for infection, but nothing else¨ (Chilean, Atacama Region, 28 years, multiparous, in a relationship).

¨only the doctor was there. A little rushed because behind me there were people that had been waiting for a long time. He chats with me breifly and says, 'I'll explain it to you someday.' I respond, 'okay'¨ (Bolivian, Atacama Region, 40 years, multiparous, single).

¨Yes, because when you go to the doctor, they don't tell you anything, they don't explain. . . they tell you it's a silent disease¨ (Bolivian, Atacama Region, 36 years, multiparous, single).

## Positive review

The positive assessment of the healthcare received after the diagnosis of Chagas disease encompasses situations that can be understood as the opposite of what was described above concerning the negative evaluations. In this way, the idea that the treatment was attentive, personalised, precise, and timely, and the perception of having received sufficient information contrasts with the lack of information and the poor manner in which it was transmitted.

It is interesting to note that, when analysing the distribution of quotes that give an account of the experiences evaluated according to the territory and nationality of the interviewees, it was observed that migrant participants and women from the regions of Tarapacá and Atacama were the ones who tended to express positive sentiments about the attention they received. This must be interpreted against the grain, as migrants are often not in the same position as national participants to express their discontent with the Chilean health system. When analysing the elements that make up a positive evaluation of the care experience among participants, the elements that emerge as defining also articulate information and attention as the dimensions that underpin the interviewees' perceptions.

Good attention is clearly defined based on the perception of having received personalised attention. In this sense, any gesture by health personnel and health institutions towards their particular needs, such as providing facilities and priority in terms of waiting times or procedures, is crucial to understanding the positive assessment of healthcare after diagnosis. For the

participants, good attention is one that considers their times, does not make them wait, is attentive to them, guides them clearly through administrative procedures, and provides timely information and containment regarding the Chagas disease.

> ¨But yes, very good attention. Because, for example, sometimes I couldn't go to get the test done, and they would arrange transportation for me, and they would ask me when I could come. So whenever I went, they were always very attentive, including the person in charge. And the secretary is always calling me. . .¨(Bolivian, Atacama Region, 46 years, multiparous, single).

> ¨Yes, I felt good, I felt comfortable (. . .) I started to feel calmer then. But with the treatment everything was fine, no problem. If I needed anything, I would go to the hospital. I would go to ask if I had any doubts, and they were always there to help, when I could. We never had any problems. When they did the tests, they would also explain them to me: that the bugs had decreased, that the treatment was going very well. . .¨ (Chilean, Atacama Region, 28 years, multiparous, in a relationship).

The adaptation of services plays a fundamental role, especially when discussing women and migrants. This is especially important when considering the complexities around work conditions. Migrants, in particular, face more significant challenges related to authorisations and the legitimacy of healthcare permits.

> ¨ as I was saying, the constancy is a seven [very good]. They are always attentive. And there are days, let's say, when I cannot ask for permission at work because it complicates things a bit for me, so she always schedules me on Mondays, when I can. So, that availability is not provided everywhere¨ (Bolivian, Metropolitan Region, 32 years, pregnant, in a relationship).

Finally, receiving timely, precise information that transmits positive messages and that they perceive as sufficient are the qualities of the last dimension that defines a positive experience.

> ¨If I needed anything, I would go to the hospital. I would go to ask if I had any doubts, and they were always there to help, when I could. We never had any problems. When they did the tests, they would also explain them to me: that the bugs had decreased, that the treatment was going very well, that we were going to cut it because I had to give my body a rest. And, with the last treatment, they said no, that we had to cut it because it was affecting my liver. So, that more than anything. Also, regarding the issue of going to the bathroom a lot, they also explained to me that it could be due to the Chagas disease, that maybe I could have it in my colon, but that that's what colonoscopy was needed for¨ (Chilean, Atacama Region, 28 years, multiparous, in a relationship).

> ¨Very good, yes. . . very good. Here, they explain things to you and also tell you what you need to do. And the attention here is excellent ¨ (Bolivian, Atacama Region, 28 years, primiparous, in a relationship).

> ¨Good, because they explained to me what it was like, uh. . . what it was about, how it was done. . . what happened to the intestine, to the heart. . . so, yeah, they explained that. And the attention is good. They already gave me a treatment, but I didn't take it, I mean. . . the tablets did me no good, I mean, the meds they gave me didn't sit well with me¨ (Bolivian, Atacama Region, 34 years, in a relationship).

Communication, providing timely information, and containing feelings of distress or unawareness are particularly significant in the care processes and, thus, in the positive evaluation of services. Communication during diagnosis is decisive for the continuity and pursuit of treatment. In cases of reaction to side effects, accompaniment, information, and communication are crucial for a successful completion of the process.

## Discussion

This study documented the elements at play in assessing healthcare experiences among women diagnosed with *Trypanosoma cruzi*, which causes Chagas disease, during their pregnancies or later. In the information dimension, the key to a positive assessment was the perception of receiving the needed knowledge. If participants had their doubts resolved and were adequately guided throughout the processes and procedures following diagnosis, they expressed satisfaction with their experience.

The assessment of the treatment was defined according to the behaviour of the health personnel with whom they interacted and the clarity in the processes and procedures they had to perform. In this way, a bad experience is defined by nasty looks, ominous tones when speaking, or the feeling that the health personnel ignored or discriminated against them. The excessive bureaucracy, the lack of clarity of procedures, the consequent waste of time, and the experiences characterised by a lack of coordination were also crucial in explaining a negatively evaluated experience. It is important to note that the care assessment focuses on three instances: undergoing the test during pregnancy, receiving the diagnosis, and undergoing the following procedures. Very few have reached treatment, and none of the interviewees has undergone a follow-up process. In addition, it is worth noting the place of bureaucracy and the lack of information concerning a transparent process or procedure associated with the expressions of the participants' discomfort.

This confirms that what is expected from the health care teams involved in applying Technical standard n˚ 162 is that they can guide and clarify regarding Chagas disease and that they are knowledgeable about the information and procedures needed to be followed after the diagnosis. In this context, the healthcare teams' understanding of the social determinants of the disease is crucial. Nevertheless, in the documentation of actions concerning Chagas disease, there is evidence of failures and ignorance regarding the social and cultural factors related to the infection, not only among those affected by the disease but also among healthcare teams [27]. The problems of otherness arise during the care of migrant users, and health institutions reproduce some aspects of cultural determinism that involve considering certain behaviours as expected only because of these women's culture [28]. In assessing healthcare experiences, there is a need for positive, clear, and tailored treatment that understands the necessities of the users but goes beyond the culturalism identified by [28]. For this reason, the creation of tools and training programs aimed at educating healthcare teams has been a vital outcome of the research. These outputs aim to improve healthcare teams' awareness of the critical role played by the ways, manners, and means of conveying accurate and clear information free from bias and prejudices. This approach to information and care for women diagnosed with Chagas disease is relevant given the prevalence shown by some research among adult and deprived women in the endemic areas in Chile [29].

The use of social grounded theory in this type of study is fundamental because it allows for collecting and identifying information never before evidenced, particularly considering the complex conditions in the regions. Despite being extensive, this research material allowed an analysis that favours the understanding of the processes related to Chagas disease in the female population that has gestated and that mostly did not know the diagnosis of Chagas. Data

collection was incredibly complex given the context of physical distance and confinement during the SARS-CoV-2 pandemic, making it necessary to adjust the study's initial characteristics and design contact strategies that would allow the interviews to be conducted. Despite this, the conversation space, given the context, was highly valued by the participants.

To conclude, it is necessary to emphasise that the products resulting from the project have been developed based on the interviews and their analysis. Futhermore, qualitative social research is indispensable for understanding social barriers, stigma, and exclusion (29). For diagnosis and its communication, the interaction with the healthcare teams is fundamental. In this interaction, communication and treatment quality play a crucial role in determining women's adherence to diagnostic and treatment processes, especially among those who postpone their healthcare to prioritize caring for their newborns and families.

## Author Contributions

**Conceptualization:** Andrea Avaria.

**Data curation:** Andrea Avaria, Camila Plaza.

**Formal analysis:** Andrea Avaria, Camila Plaza.

**Funding acquisition:** Andrea Avaria.

**Investigation:** Andrea Avaria.

**Methodology:** Andrea Avaria.

**Project administration:** Andrea Avaria.

**Resources:** Andrea Avaria.

**Software:** Camila Plaza.

**Supervision:** Andrea Avaria.

**Validation:** Andrea Avaria.

**Visualization:** Andrea Avaria.

**Writing – original draft:** Andrea Avaria, Camila Plaza.

**Writing – review & editing:** Andrea Avaria, Camila Plaza.

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
