## [Decision Letter · Decision Letter 0]

9 Jul 2024

PONE-D-24-17871Health Experiences of Pregnant Women with Chagas Disease in the Atacama, Tarapacá, and Metropolitan Regions of Chile. (Poor) Treatment as an Indicator of Healthcare Barriers.PLOS ONE

Dear Dr. Avaria,

Thank you for submitting your manuscript to PLOS ONE. After careful consideration, we feel that it has merit but does not fully meet PLOS ONE’s publication criteria as it currently stands. Therefore, we invite you to submit a revised version of the manuscript that addresses the points raised during the review process.

Dear Authors,

Please, follow all reviewers' suggestions. I strongly suggest to change the title as not all study participants were pregnant at the time of the interview and most of them were not born in Chile. Other aspect of the title is “poor treatment”. The reader will think that the authors will describe the frequency of previous treatment with benznidazole or nifurtimox the patients had. But that was not the case. So, I also suggest to change the word “treatment” in the title.  

We look forward to receiving your revised manuscript.

Kind regards,

Roberto Magalhães Saraiva, MD, PhD

Academic Editor

PLOS ONE

2. Please provide additional details regarding participant consent. In the ethics statement in the Methods and online submission information, please ensure that you have specified what type of consent you obtained (for instance, written or verbal, and if verbal, how it was documented and witnessed). If your study included minors, state whether you obtained consent from parents or guardians. If the need for consent was waived by the ethics committee, please include this information.

3. n the online submission form you indicate that your data is not available for proprietary reasons and have provided a contact point for accessing this data. Please note that your current contact point is a co-author on this manuscript. According to our Data Policy, the contact point must not be an author on the manuscript and must be an institutional contact, ideally not an individual. Please revise your data statement to a non-author institutional point of contact, such as a data access or ethics committee, and send this to us via return email. Please also include contact information for the third party organization, and please include the full citation of where the data can be found.

Additional Editor Comments:

Dear Authors,

Please, follow all reviewers' suggestions. I strongly suggest to change the title as not all study participants were pregnant at the time of the interview and most of them were not born in Chile. Other aspect of the title is “poor treatment”. The reader will think that the authors will describe the frequency of previous treatment with benznidazole or nifurtimox the patients had. But that was not the case. So, I also suggest to change the word “treatment” in the title.

Reviewers' comments:

Reviewer's Responses to Questions

**Comments to the Author**

1. Is the manuscript technically sound, and do the data support the conclusions?

Reviewer #1: Yes

Reviewer #2: Yes

2. Has the statistical analysis been performed appropriately and rigorously? 

Reviewer #1: N/A

Reviewer #2: N/A

3. Have the authors made all data underlying the findings in their manuscript fully available?

Reviewer #1: Yes

Reviewer #2: Yes

4. Is the manuscript presented in an intelligible fashion and written in standard English?

Reviewer #1: Yes

Reviewer #2: Yes

5. Review Comments to the Author

Reviewer #1: Dear Editor and Authors,

Thank you for the opportunity to review the manuscript titled "Health Experiences of Pregnant Women with Chagas Disease in the Atacama, Tarapacá, and Metropolitan Regions of Chile: Treatment as an Indicator of Healthcare Barriers". The paper provides valuable insights into healthcare system perceptions regarding Chagas disease among Chilean and migrant pregnant women. I found the manuscript highly informative. The detailed testimonials are crucial for guiding healthcare policies not only in Chile but also in other countries facing Chagas disease as a public health challenge. Therefore, I recommend the publication of the manuscript after minor revisions:

• Line 12. Replace “forms of Chagas transmission worldwide” with “forms of Trypanosoma cruzi transmission worldwide”. Clarify throughout the text that it is the parasite, not the disease itself, that is transmitted.

• Throughout the manuscript: Correctly refer to "Chagas disease" instead of just "Chagas", the name of the discoverer.

• Line 16. Italianize “Trypanosoma cruzi” consistently throughout the manuscript.

• Line 39. Clarify whether Chile is free of vector transmission caused by domiciled species only (e.g., Triatoma infestans) or all Triatominae species. Consider discussing potential vectorial transmission by Mepraia spp.

• Line 71. Ensure reference [20] is correctly placed at the end of the sentence.

• Line 127. Include median age and interquartile range.

• Line 128. Specify the other two South American countries with similar participant characteristics due to the small sample size.

• Line 128 (and others). Spell out numbers below thirteen for consistency.

• Line 126-131. Consider presenting these results in a table format for clarity.

• Line 133. Use "Chagas" with capitalization throughout the manuscript, honoring Dr. Carlos Justiniano Ribeiro Chagas.

• Line 151. Remove the unnecessary repetition of "as".

• Line 268: Insert a space between "repeated" and "across".

• Line 362. Clarify "standard 162" as it appears to be a typographical error.

Reviewer #2: Thank you for the work you have done in Chile to bring attention to migrant women who are living with Chagas disease who are either pregnant or postpartum. The manuscript is otherwise written well, and results are easy to interpret. I have some additional suggestions to help improve the manuscript as it stands and prior to publication:

- Please review for grammatical errors. Please be consistent when discussing "Chagas" and would recommend either leaving as "Chagas disease" throughout manuscript or abbreviate as "CD".

I would consider how you use "vertical transmission" and "vertical Chagas". Many people get confused between vertical and vectorial transmission as the terms are very similar appearing. "Congenital Chagas disease" is a better term to be used when discussing the disease itself. I recommend adding "congenital Chagas disease" to keywords

- in the abstract on line 15-16, "carriers of Trypanosoma cruzi" is not a great way to describe someone living with Chagas disease. I would re-word this sentence and possibly utilize "living with Chagas disease"

- I suggest adding a figure that contains a map of the three regions where the study was conducted. Within the map, I would also suggest adding population density and catchment. What is the "Metropolitan Region"? More clarification on the sites is important. You can add the clinics to the map.

- please clarify recruitment of the patients in lines 99-102. Which healthcare institutions? I would add participating clinical institutions to the "acknowledgements".

- Line 100: diagnosed with "Chagas disease"

- Did the team assess if the participants had concerns for transmission in Chile, Bolivia or other regions of south America? Where did the other two migrant from in South America? Was there concern for congenital Chagas in the participants or other where they coming from regions of vectorial transmission?

- The title needs attention. After reading the manuscript it appears that many of the participants were "migrants" and I would suggest adding to the title somehow. There is a major focus throughout the study on migratory health and I think it should be added.

- another question about the title is "pregnant" women. It appears that interviews occurred during pregnancy and postpartum periods. May consider adding "Pregnant and Postpartum Women..."

- I would consider discussing stigma in those living with Chagas. https://www.ncbi.nlm.nih.gov/pmc/articles/PMC3772024/

https://academic.oup.com/trstmh/article/114/7/476/5734980

6. PLOS authors have the option to publish the peer review history of their article (what does this mean?). If published, this will include your full peer review and any attached files.

Reviewer #1: **Yes: **Fred Luciano Neves Santos

Reviewer #2: **Yes: **Norman L. Beatty, University of Florida College of Medicine

---

## [Author Response · Author response to Decision Letter 0]

2 Oct 2024

Response to Reviewers. September 10, 2024.

Dear Reviewers,

We sincerely appreciate the review, as well as your comments and suggestions. We hope that our revisions have enriched the text accordingly. We have incorporated most of your recommendations, which contribute to a better understanding of the document.

We have attached the revised article file with track changes enabled to highlight the modifications made. 

Below, we respond to each of the reviewers' and editors' requests. We have copied each recommendation and provided our response beneath it, indicated with an "R." (in red)

We would like to express our gratitude once again for the valuable comments and suggestions provided by the reviewers. Andrea and Camila

Editors: 

1. Title changes requested: The title has been modified as per the request. 

2. Regarding informed consent: It has been clarified that informed consent was obtained in written form pre-pandemic, and verbally during and after the pandemic, with the record of this information preserved in the audio of the interviews.

3. The email of the Ethics Committee has been provided as the contact point to receive any requests for the data if needed. 

4. The title of the online form has been changed. 

Revisor #1:

1. Line 12. Replace “forms of Chagas transmission worldwide” with “forms of Trypanosoma cruzi transmission worldwide”. Clarify throughout the text that it is the parasite, not the disease itself, that is transmitted. 

R: Line 12 has been revised and changed to “forms Trypanosoma cruzi transmission worldwide.”

2. Throughout the manuscript: Correctly refer to "Chagas disease" instead of just "Chagas", the name of the discoverer.

R: I have corrected the term “Chagas” to “Chagas disease” throughout the text. However, I have used the term “Chagas problem” when discussing the disease from a broader perspective related to the implications of carrying Trypanosoma cruzi without having developed Chagas disease. 

3. Line 16. Italianize “Trypanosoma cruzi” consistently throughout the manuscript.

R: Trypanosoma cruzi has been italicized throughout the text. 

4. Line 39. Clarify whether Chile is free of vector transmission caused by domiciled species only (e.g., Triatoma infestans) or all Triatominae species. Consider discussing potential vectorial transmission by Mepraia spp.

R: The issue regarding the significant Triatoma species in Chile, relevant to the transmission of T. cruzi, has been clarified in the text. 

According to the objective of the national vector control program, the focus has been on controlling the native domiciliary vector, Triatoma infestans, a species of epidemiological importance in Chile and the southern cone. In this context, the vector transmission of T. cruzi through Triatoma infestans has been interrupted. In the national context, there are no records of evidence of T. cruzi transmission through wild vectors.

5. Line 71. Ensure reference [20] is correctly placed at the end of the sentence.

R: The reference on Line 71 has been correctly placed at the end of the sentence.

6. Line 127. Include median age and interquartile range. 

R. Given the characteristics of the study, I do not consider these data significant enough to incorporate. But the median is 32. And the Q1 : 28 and Q3 36 the RIC is 8.0 (36-28)

7. Line 128. Specify the other two South American countries with similar participant characteristics due to the small sample size.

R: This information has been incorporated.

8. Line 128 (and others). Spell out numbers below thirteen for consistency.

R: All numbers have been kept in numerical format. 

9. Line 126-131. Consider presenting these results in a table format for clarity.

R. It is considered that presenting the data in text format is clearer than using a table, as the highlighted information relates to various topics, making it difficult to consolidate all the information in a single table.

10. Line 133. Use "Chagas" with capitalization throughout the manuscript, honoring Dr. Carlos Justiniano Ribeiro Chagas.

R. Capitalization has been applied throughout the text when referencing "Chagas.". 

11. Line 151. Remove the unnecessary repetition of "as".

R. It has been removed. 

• Line 268: Insert a space between "repeated" and "across".

R. A space has been added.

• Line 362. Clarify "standard 162" as it appears to be a typographical error.

R: It has been revised and corrected. Technical Standard No. 162 refers to the technical guidelines and normative issued by the Ministry of Health to address the problem of Chagas disease at the national level.

Reviewer #2: 

1.Thank you for the work you have done in Chile to bring attention to migrant women who are living with Chagas disease who are either pregnant or postpartum. The manuscript is otherwise written well, and results are easy to interpret. I have some additional suggestions to help improve the manuscript as it stands and prior to publication:

- Please review for grammatical errors. Please be consistent when discussing "Chagas" and would recommend either leaving as "Chagas disease" throughout manuscript or abbreviate as "CD".

R: We have reviewed the manuscript for grammatical errors and corrected those detected.

We have changed "Chagas" to "Chagas disease" where appropriate. We used "Chagas issue" in a few instances to refer to it from a broader perspective. In the original manuscript, we used the concept of "Chagas" with the understanding that it represents a complex and multidimensional issue that affects both the individual and the community, extending beyond the mere development of the disease. Nevertheless, we have decided to take your suggestion into account and use "Chagas disease" throughout most of the manuscript, except in the parts mentioned above.

2. I would consider how you use "vertical transmission" and "vertical Chagas". Many people get confused between vertical and vectorial transmission as the terms are very similar appearing. "Congenital Chagas disease" is a better term to be used when discussing the disease itself. I recommend adding "congenital Chagas disease" to keywords

R. We have corrected the text to refer to “congenital Chagas disease” to avoid any potential confusion.

3. in the abstract on line 15-16, "carriers of Trypanosoma cruzi" is not a great way to describe someone living with Chagas disease. I would re-word this sentence and possibly utilize "living with Chagas disease"

R. We have rephrased that paragraph. 

4. I suggest adding a figure that contains a map of the three regions where the study was conducted. Within the map, I would also suggest adding population density and catchment. What is the "Metropolitan Region"? More clarification on the sites is important. You can add the clinics to the map.

R: We have incorporated a map that shows the location of the three regions and includes the most significant data related to the diagnosis of pregnant women in those regions, as well as data on migrants in each region and at the national level. 

5. please clarify recruitment of the patients in lines 99-102. Which healthcare institutions? I would add participating clinical institutions to the "acknowledgements".

R: We have rephrased the text as follows: The participants were selected by the treating healthcare teams at the hospital or primary care level in each region. These teams facilitated the first contact with the women. Following this, the local project teams contacted the women by phone, confirmed their participation, and obtained informed consent. All participants had to have been diagnosed positive for Chagas disease after 2016, when Technical Standard n°162 was implemented in the Chilean health system.

We would like to thank the Chilean Ministry of Health and the Seremi, health services, reference hospitals, and Cesfam centers of the Atacama, Tarapacá, and the Metropolitan regions.

6. Line 100: diagnosed with "Chagas disease"

R: Corrected 

7. - Did the team assess if the participants had concerns for transmission in Chile, Bolivia or other regions of south America? Where did the other two migrant from in South America? Was there concern for congenital Chagas in the participants or other where they coming from regions of vectorial transmission?

R: The majority of the women interviewed were from Bolivia, Chile, Argentina and Venezuela. The interviewing team identified concerns related to information on congenital Chagas disease and barriers associated with health communication. 

8. The title needs attention. After reading the manuscript it appears that many of the participants were "migrants" and I would suggest adding to the title somehow. There is a major focus throughout the study on migratory health and I think it should be added.

R: The title has been changed. 

9. another question about the title is "pregnant" women. It appears that interviews occurred during pregnancy and postpartum periods. May consider adding "Pregnant and Postpartum Women..."

R: It has been changed in the title and in the text. 

10. I would consider discussing stigma in those living with Chagas. https://www.ncbi.nlm.nih.gov/pmc/articles/PMC3772024/

https://academic.oup.com/trstmh/article/114/7/476/5734980

R: The text has been modified to include the contribution of qualitative social research in better understanding the social barriers, stigma, and exclusion related to Chagas disease, as well as the importance of addressing the social components that affect individuals living with this disease.

Response to the query stated in the email LOS ONE: Edits requested on your submission PONE-D-24-17871R1 - [EMID:eb694cc9daf01626: 

Q2. We understand that your data availability statement currently reads as follows: "Data cannot be shared publicly because of Fonis requirement. Data are available from the FONIS, ANID Chile. Ethics Committee (contact via comite.etica@uautonoma.cl) for researchers who meet the criteria for access to confidential data."

Are these third party data (i.e., data not owned or collected by the author(s)) belonging to FONIS? If these are indeed third party data, please confirm that others would be able to access these data in the same manner as the authors and that the authors did not have any special access privileges that others would not have.

Response

The National Fund for Health Research (FONIS) finances applied research projects in health. This research received this national funding. 

Fonis does not have ownership over the data, it ensures that it is public and open access. 

As part of the commitments made with the Ministry of Science of Chile, through the FONIS, the products must be open access and public, which is why we have a website associated with the www.chaochagaschile.cl project, where the materials produced, and the results through the various publications are available for use and download. However, neither the fund (FONIS) nor the university has an open repository for the data collected, these are in the hands of the person responsible for the research. In this case, the Ethics Committee of the Universidad Autónoma de Chile, as the institution responsible for the execution of the project, undertakes to make the results available to those who request them in accordance with access criteria that are consistent with the principles of public use and common good. Contact via comite.etica@uautonoma.cl

That is to say, the Ethics Committee of the University is committed and becomes part of it, assumes the responsibility of providing the information that any person requires, regarding the qualitative data collected by the Researcher Andrea Avaria, through the project financed by the Ministry of Science, with funds from FONIS.

---

## [Decision Letter · Decision Letter 1]

25 Oct 2024

Health Experiences of Pregnant and Women with Chagas Disease in the Atacama, Tarapaca, and Metropolitan Regions of Chile. Mistreatment as an Indicator of Healthcare Barriers.

PONE-D-24-17871R1

Dear Dr. Avaria,

We’re pleased to inform you that your manuscript has been judged scientifically suitable for publication and will be formally accepted for publication once it meets all outstanding technical requirements.

Kind regards,

Roberto Magalhães Saraiva, MD, PhD

Academic Editor

PLOS ONE

Additional Editor Comments (optional):

Reviewers' comments:

Reviewer's Responses to Questions

**Comments to the Author**

1. If the authors have adequately addressed your comments raised in a previous round of review and you feel that this manuscript is now acceptable for publication, you may indicate that here to bypass the “Comments to the Author” section, enter your conflict of interest statement in the “Confidential to Editor” section, and submit your "Accept" recommendation.

Reviewer #1: All comments have been addressed

2. Is the manuscript technically sound, and do the data support the conclusions?

Reviewer #1: (No Response)

3. Has the statistical analysis been performed appropriately and rigorously? 

Reviewer #1: (No Response)

4. Have the authors made all data underlying the findings in their manuscript fully available?

Reviewer #1: (No Response)

5. Is the manuscript presented in an intelligible fashion and written in standard English?

Reviewer #1: (No Response)

6. Review Comments to the Author

Reviewer #1: (No Response)

7. PLOS authors have the option to publish the peer review history of their article (what does this mean?). If published, this will include your full peer review and any attached files.

Reviewer #1: **Yes: **Fred Luciano Neves Santos

---

## [Editor Report · Acceptance letter]

30 Oct 2024

PONE-D-24-17871R1 

PLOS ONE

Dear Dr. Avaria, 

I'm pleased to inform you that your manuscript has been deemed suitable for publication in PLOS ONE. Congratulations! Your manuscript is now being handed over to our production team.

Kind regards, 

on behalf of

Dr. Roberto Magalhães Saraiva 

Academic Editor

PLOS ONE